# Tumor-Associated Protein Profiles in Kaposi Sarcoma and Mimicking Vascular Tumors, and Their Pathological Implications

**DOI:** 10.3390/ijms20133142

**Published:** 2019-06-27

**Authors:** Kyoung Bun Lee, Kyu Sang Lee, Hye Seung Lee

**Affiliations:** 1Department of Pathology, Seoul National University Hospital, Seoul 110-799, Korea; 2Department of Pathology, Seoul National University Bundang Hospital, Seongnam 463-707, Korea

**Keywords:** Kaposi sarcoma, angiosarcoma, hemangioendothelioma, hemangioma

## Abstract

We investigated protein profiles specific to vascular lesions mimicking Kaposi sarcoma (KS), based on stepwise morphogenesis progression of KS. We surveyed 26 tumor-associated proteins in 130 cases, comprising 39 benign vascular lesions (BG), 14 hemangioendotheliomas (HE), 37 KS, and 40 angiosarcomas (AS), by immunohistochemistry. The dominant proteins in KS were HHV8, lymphatic markers, Rb, phosphorylated Rb, VEGF, and galectin-3. Aberrant expression of p53, inactivation of cell cycle inhibitors, loss of beta-catenin, and increased VEGFR1 were more frequent in AS. HE had the lowest Ki-67 index, and the inactivation rates of cell cycle inhibitors in HE were between those of AS and BG/KS. Protein expression patterns in BG and KS were similar. Clustering analysis showed that the 130 cases were divided into three clusters: AS-rich, BG-rich, and KS-rich clusters. The AS-rich cluster was characterized by high caveolin-1 positivity, abnormal p53, high Ki-67 index, and inactivated p27. The KS-rich cluster shared the features of KS, and the BG-rich group had high positive expression rates of galectin-3 and low bcl2 expression. In conclusion, although the rate was different, AS and HE tended to have less cell cycle marker expression than KS, and features of BG and activated KS cell signaling were similar.

## 1. Introduction

Kaposi sarcoma (KS) is an intermediate grade, rarely metastasizing vascular tumor caused by human herpes virus 8 (HHV8) infection [1]. According to the epidemiology data on HHV8, four types of KS exist; however, in Korea, KS generally occurs in immunocompromised or AIDS patients. KS has three stages: patched, plaque, and nodular, and these stages reflect not only the different clinical manifestations but also histological differences. The patched stage is an early stage characterized by small, irregularly dilatated vessels in the edematous dermis that are mixed with inflammatory cells or extravasated red blood cells (RBCs). At this stage, KS has similar histologic features as BG such as stasis dermatitis, benign lymphangioma, targetoid hemosiderotic hemangioma, immature scar, or pyogenic granuloma [2]. As KS advances, the cellular component increases, the vessel-like feature of the early stage disappears, and spindle cells are the main component mixed with some inflammatory cells and extravasated RBCs. At this stage, mesenchymal tumors in skin and subcutaneous tissue, such as angiosarcoma (AS), hemangioendothelioma (HE), aneurysmal fibrous histiocytoma, spindle cell hemangioma, and dermatofibrosarcoma protuberance, should be differentiated [2]. After the establishment of the pathogenic role of HHV8 in KS, the presence of HHV8 in tissues became the essential diagnostic criterion for KS. Because the monoclonal antibody for HHV-latent nuclear antigen-1 (LNA-1) has been frequently used on formalin-fixed paraffin tissue, diagnosis of KS in a pathology laboratory is no longer challenging [2]. Recently, many researchers showed a role for HHV8 in the pathophysiology of KS’s stepwise progression and morphogenesis [1]. Viral proteins disturb cell cycle proteins to activate cellular proliferation, turn infected cells into lymphatic lineage cells, and evoke inflammatory reactions in surrounding tissues [1]. This mechanism provides a useful model for the understanding of the pathogenesis of vascular tumors and their biological features. 

Vascular tumors in skin and soft tissues are generally classified into four categories: benign; intermediate, locally aggressive; intermediate, rarely metastasizing; and malignant lesions, according to their clinical behavior and morphological features [3]. Although each entity has specific clinicopathological features, atypical vascular lesions with insufficient clinical information or unusual clinical settings may make the pathology difficult to diagnose in clinic. Benign vascular lesions (BG) can sometimes look like high-grade vascular tumors, especially after recurring irritation. On the other hand, the peripheral portion of AS has minimally increased vascular channels, and nuclear atypia is so subtle that it can be easily overlooked by pathologists.

In the present study, we aimed to determine specific protein profiles in benign, intermediate, and malignant vascular tumors by assessing the expression of 26 proteins. Using this method, we aimed to enable the differential diagnosis of vascular lesions, and compare the differences and similarities in pathophysiology between each tumor group and KS. 

## 2. Results

### 2.1. Exclusive Expression of HHV8 and Lymphatic Differentiation of KS

Table 1 shows the summary of the expression rates for 26 markers in the four disease groups, and Figure 1a demonstrates the percentage of abnormal expression as a radar chart. HHV8 was exclusively expressed in KS (100% in KS vs. 0% in BG, HE, and AS, *p* < 0.001). D2-40 and podoplanin expression rates were also higher in KS compared to those in other groups (100% and 89%, respectively, in KS vs. 21%–48% and 10%–33%, respectively, in BG, HE, and AS, *p* < 0.001). Positive rates of CD31 and CD34 were relatively high in all groups, but CD34 expression was lower in AS compared with that in other groups (73% in AS vs. 91%–100% in BG, HE, and KS, *p* < 0.001). With the exception of KS, each group expressed D2-40 and podoplanin in some areas, implying lymphatic differentiation. However, there were no significant differences in expression of these proteins between BG, HE, and AS. Figure 1b shows a cross-table summarizing pairwise comparisons within the four groups.

### 2.2. Low Proliferative Activity in HE

Proliferative activity assessed using the Ki-67 index was lowest in the HE group, compared to that of the other three groups (Table 1). The mean Ki-67 index in the HE group (5 ± 5.9%, mean ± SD) was significantly lower than that of the BG group (23.2 ± 22.1%, *p* = 0.008) and AS group (33.4 ± 28.5, *p* < 0.001). Interestingly, the BG and AS groups showed similar high Ki-67 index levels (*p* < 0.091). The Ki-67 index of the KS group (19.5 ± 18.3) was in between that of the HE and BG/AS groups, although there was no statistical significance (Figure 1b).

Benign vascular lesions with Ki-67 expression exceeding the median value were capillary hemangioma (3/10) and granuloma pyogenicum (12/18); Ki-67 expression levels in these lesions were 46.33% ± 16.84% (mean ± SD). BG lesions with Ki-67 expression below the median value included acroangiodermatitis, angiofibroma, cavernous hemangioma, cherry angioma, hemangioma, intravascular histiocytosis, and stasis dermatitis; Ki-67 expression levels were 8.71% ± 8.10% (mean ± SD).

### 2.3. Activation of Rb Signaling in KS, and Inactivation of Cell Cycle Inhibitors and Aberrant p53 in AS

Among 10 proteins involved in cell cycle, apoptosis, and cell survival, Rb and phosphorylated Rb (pRb) were more highly expressed in the KS group (97% and 92%, respectively), when compared to their expression in BG, HE, and AS (50%–92% and 62%–68%, *p* < 0.001 and *p* = 0.009, respectively, Table 1 and Figure 1a). Among cyclin-dependent kinase inhibitors, p16 and p27 were more frequently inactivated in AS (50% and 60%, respectively) than in BG (21% and 15%, *p* = 0.012 and *p* < 0.001, respectively). The rates of inactivated p27 in the HE group were also higher than those in the BG and KS groups (57% in HE vs. 15%–22% in BG and KS, *p* = 0.017 for HE vs. BG, and *p* = 0.04 for HE vs. KS). Aberrant p53 expression (implying abnormal p53 status) was found only in the AS and HE groups, with no positive KS and BG cases. Positive expression of cell survival markers was highest in AS and lowest in HE. The BG and KS groups showed medium levels of cell survival marker expression that were between those of the AS and HE groups. Bcl2 and NF-κB were more upregulated in KS than in AS. Positive expression rates and p-values are listed in detail in Table 1.

### 2.4. Expression of VEGFR1 and C-Kit in AS

Out of the four tumor groups, AS had the highest positive expression of VEGFR1 (33% in AS vs. 0–11% in other groups, *p* = 0.003). C-kit was exclusively expressed in AS (4/40 [10%]). However, expression of VEGF was slightly higher in the KS and AS groups than in BG and HE, but this difference was not significant (43% and 49% vs. 21% and 14%, respectively, *p* = 0.063).

### 2.5. Inverse Correlation of Galectin-3 and Caveolin-1 Expression between BG/KS and AS/HE

Comparative analysis of cell adhesion and motility-associated proteins showed that positive rates of galectin-3 in BG and KS groups (67% and 76%, respectively) were higher than those in HE and AS groups (36% and 38%, respectively, *p* < 0.001). Conversely, caveolin-1 expression was higher in HE and AS groups (71% and 70%, respectively) than in BG and KS (36% and 35%, respectively, *p* = 0.005). Other markers that showed differential expression in specific tumor groups included CD44, EMA, and β-catenin. CD44 was uniquely downregulated in KS (16% in KS vs. 33%–36% in BG, HE, and AS, all pairwise *p* < 0.05). Positive expression of EMA was uniquely found in AS (4/40 [10%]), and total loss of β-catenin expression was more pronounced AS (15%) than in the other groups (0–7% in BG, HE, and AS, *p* = 0.033).

### 2.6. Differential Power and Protein Profiles of Hierarchical Clusters

We tried to classify disease groups based on 24 protein profiles (and excluding HHV8). Hierarchical clustering showed three large clusters (Figure 2a). The distribution of disease groups in each cluster is charted in Figure 2b. Cluster 1 comprised 55% of the AS group, cluster 2 comprised 62% of the BG group, and cluster 3 comprised 95% of the KS group. Unusual expression patterns of markers in each cluster identified by pairwise comparisons between clusters are listed in Table 2 and shown in Figure 2c. Cluster 1 was characterized by 1) high expression of caveolin-1, expression of p53, high Ki-67 index, and expression of c-kit, 2) frequent inactivation of p27, and 3) low expression of galectin-3 and CD34. Cluster 2 was characterized by 1) high expression of CD44 and 2) low expression of p21, VEGF, and bcl2. Finally, cluster 3 was characterized by 1) high expression of HHV8, D2-40, podoplanin, Rb and pRb and 2) low levels of inactivated p16 or loss of β-catenin. The remaining seven markers, including E-cadherin, Jagged-1, EMA, VEGFR1, cyclinD1, cell survival markers, and CD31, did not show significant differences between clusters.

## 3. Discussion

The 2013 World Health Organization (WHO) classification of vascular tumors in soft tissue and bone comprises four groups of vascular tumors: benign, locally aggressive intermediate, rarely metastasizing intermediate, and malignant tumors [3]. The 2018 WHO classification of skin tumors included vascular tumors under soft tissue tumors of the skin, following the disease scheme and international classification of diseases for oncology (ICD-O) of soft tissue [4]. Hemangioma is the main component of benign lesions; however, depending on the clinical setting and manifestations, reactive vascular proliferative lesions with dermatitis can also be classed as BG in skin. Intermediate-grade vascular tumors are called HE, which comprise kaposiform HE, retiform HE, papillary intralymphatic angioendothelioma, composite HE, epithelioid-sarcoma-like HE, and KS [3]. This group of tumors can repeatedly recur and rarely metastasizes to other organs. AS and epithelioid HE are both malignant vascular tumors with poor prognosis. Despite different prognosis and morphology, gray-zone areas between these disease types makes them difficult to diagnose. In the present study, we assessed the expression status of 26 tumor-associated proteins in 130 vascular tumors classed as four diagnostic groups—BG, HE, KS, and AS—and performed bidirectional comparative analysis. Firstly, we performed a direct comparison between groups and then performed a clustering analysis based on the similarities in protein expression. Using this method, we narrowed down differently expressed markers to identify unique protein profile patterns for each diagnostic group. KS was characterized by the presence of HHV8, lymphatic differentiation, and activated Rb signaling (including pRb). These findings are compatible with HHV8’s role in the pathogenesis of KS. LNA-1 is one of the viral proteins expressed in the viral latent phase, which inhibits p53 and modulates pRb and GSK3β for the transition to G1-S [5]. Viral homologous of cellular cyclin D, vcyc, reacts with cyclin-dependent kinase and is not inhibited by common cell cycle inhibitors [6,7]. Lymphatic differentiation in KS is a well-known distinctive feature of this tumor type (even though it is absent in lymphangioma and kaposiform HE) [8] and occurs due to the regulation of Prox-1, a key regulator of lymphatic differentiation, by HHV8 [7]. In contrast with the KS group, the main feature of AS observed in the present study was aberrant expression of p53, inactivation of p16 and p27, loss of β-catenin, and increased VEGFR1. Abnormal p53 signaling has been reported in hepatic AS, and inactivated p16 was shown to induce endothelial cell dysfunction that led to AS in an in vitro study [9,10]. Upregulation of VEGFR1 (also known as FLT1) and other vascular-specific receptor tyrosine kinases, including TIE1, KDR, SNRK, and TEK, has also been previously reported in AS [11]. C-kit expression in AS has been reported to be higher in soft tissue AS than in bone AS [12]. In our study, four cases of AS were positive for c-kit in skin, liver, breast, and heart tissues [13]. 

HE was characterized by having the lowest Ki-67 index in the four groups, which was a difference large enough to enable differentiation between HE and the other tumor types. This feature can also be helpful to differentiate AS or BG tumors. However, the use of the Ki-67 index for differentiation between benign and vascular tumors is controversial [2,14]. In our study, the Ki-67 index in BG was similar to that of the AS group, and higher than that of HE. The high proliferative activity in BG seems to be analogous to that of KS, which occurs via the Rb pathway and is induced by viral proteins. The triggers for cellular proliferation in BG are hypoxia, inflammation, or genetic mutations [15,16]. In our study, HE showed similar protein profile patterns as those in AS, but aberrant expression was generally lower in HE than in AS. These results may imply the sequential progression from HE to AS. Interestingly, epithelioid HE is considered to be an indolent tumor, but 20–30% of these tumors can rapidly progress and show histological features identical to those seen in conventional AS [17]. Following clustering analysis, the HE group did not form a dominant cluster, even though 50% of HE was clustered with the AS-rich cluster 1. This may offer further evidence of a sequential step between HE and AS.

Caveolin-1 and galectin-3 showed contrasting expression levels in AS/HE and BG/KS. Galectin-3 was more frequently expressed in BG and KS groups, whereas caveolin-1 showed the opposite. Caveolin-1 is a scaffolding protein found in the plasma membrane, which mediates transport of intracellular and extracellular proteins and signal transduction. Differential expression levels of caveolin-1 have been correlated with myogenic differentiation of rhabdomyosarcoma [18]. Galectin-3 is a member of the lectin family and plays a role in cell-to-cell adhesion, cell-matrix interaction, and cancer metastasis [18]. In the metastatic model, galectin-3 is expression in vascular endothelial cells, and its free form is increased in serum [19]. However, the role of galectin-3 in vascular tumors or other soft tissue tumors is unknown. Importantly, the synergistic or coordinating interaction between galectin-3 and caveolin-1 has been reported to promote tumor cell migration and invasion in thyroid cancer [20].

CD44 is a blood vascular endothelium-specific hyaluronan receptor [21]. It is unclear whether CD44 is downregulated in KS during virus-induced lymphatic programing or KS arises from HHV8-infected lymphatic endothelial cells (LEC) not expressing CD44. Another hypothesis is that stem cells are involved in the pathogenesis of KS. CD44 is one of several surface markers for mesenchymal stem cells, and HHV8-infected mesenchymal stem cells show a loss of stem cell markers [22].

Cluster analysis was used to investigate specific characteristics in each group. Out of the 130 cases, 62% (81) were successfully classed into three clusters. However, the remaining 38% cases were distributed within other clusters. The most accurate clustering was that of the KS group (98%), whereby only two cases were misplaced. We tried to determine the clinicopathologic difference between clustered cases and non-clustered cases (for example, AS in AS-rich cluster 1 vs. AS in BG-rich cluster 2), but the small size of groups was a limitation for this analysis. Interestingly, the cluster pattern and protein expression differed according to tumor location (Appendix A). Five cases of AS in KS-rich cluster 3 were skin lesions and correlated with high expression of D2-40 and podoplanin, suggesting a lymphatic phenotype. However, NF-κB and galectin-3 were more frequently expressed in skin HE than in non-skin HE. Calveolin levels and loss of p16 were lower in skin lesions than in non-skin lesion. These results confirmed the biologic differences between skin AS and non-skin AS and suggested the biologic heterogeneity of HE according to location.

In conclusion, we were able to find patterns that differentiated BG, HE, KS, and AS by comparing the expression of 26 tumor-associated proteins. BG and KS shared a “hyperplastic pattern” marked by activated proliferative features, even though the underlying pathogen was different. HE and AS shared aberrant expression of cell cycle inhibitors, despite the differences in their tumor grade. Thus, vascular tumors mimicking Kaposi sarcoma can be categorized into three clusters: cluster 1, altered expression of tumor suppressor genes (p53, p27) with a high proliferative index, enriched AS; cluster 2: high CD44 expression with a low proliferative index suggesting stem cell differentiation, mixed diagnosis; and cluster 3: lymphatic differentiation with activated Rb pathway, enriched KS.

## 4. Materials and Methods

### 4.1. Patients

We selected vascular neoplasm patients within surgical pathology archives, covering the period between 1999 and 2009, from the Seoul National University Hospital and Seoul National University Bundang Hospital. The BG group comprised only patients with skin lesions, whereas the HE, KS, and AS patients were selected with no site limitation due to low incidence rates, compared to those of BG. We retrospectively enrolled 130 cases from a total of 107 patients, including 39 BG cases/patients, 14 HE cases from 11 patients, 37 KS cases from 26 patients, and 40 AS cases from 28 patients. For all cases included in the study, medical records and paraffin block tissue samples were obtained. The detailed pathological diagnosis of each study group is listed in Table 3. Age, sex, sampling method, site, immune status, disease progression, and survival were obtained from medical and pathologic records. Disease progression was defined as recurrent lesions in previously treated sites, tumor growth in untreated lesions, or new lesions in other sites confirmed by clinical examination or radiology. Demographic data of the 107 patients is summarized in Table 4. The average ratio of male to female was 1.7:1 in all patient groups, and this ratio was highest in the KS group (6.3:1). The median age of patients was 62 years old, with patients from the HE group being the youngest (4.5 years). The ratio of resected cases was highest in the AS group (79% in AS vs. 31% in all groups). In the BG and KS groups, the portion of skin lesioned was higher than in the HE and AS groups (100% in BG and 86% in KS vs. 55% in HE and 46% in AS). The number of immune-compromised patients was highest in KS (69% in KS vs. 10–18% in BG, HE, and AS). Disease progression and death due to cancer were highest in AS and lowest in BG (57% in AS vs. 3% in BG). This study was approved by the Institutional Review Board for human subjects’ research of the Seoul National University Hospital (IRB No. 1211-080-442).

### 4.2. Immunohistochemical Staining

Four-µm-thick tissue sections on glass slides were deparaffinized at 60 °C for 1 hour, dewaxed at 72 °C for 3 minutes, and hydrated by washing with alcohol three times. Immunohistochemical staining was performed using two types of autostainer, Leica Bond-max autostainer (Leica Biosystems, Melbourne, Australia) and Ventana BenchMark XT staining system (Ventana Medical Systems, Inc., Tucson, AZ, USA). Antigens were retrieved by heat treatment in the epitope retrieval solution for each autostainer (pH 6.0 Bond epitope retrieval solution 1, pH 9.0 Bond epitope retrieval solution 2, or Ventana mild 32 condition reagent). After peroxidase blocking for 5 minutes, the primary antibody was incubated for 15 minutes. The type, dilution level, and company information of the 25 primary antibodies used are listed in Table 5. After the primary antibody reaction, a chromogenic procedure was done using the Bond polymer detection kit (Leica Biosystems, Cat. No. DS9800) or Ventana ultraView universal DAB detection kit (Ventana Medical Systems, Cat. No. 760-500). Counterstaining for nuclei was done by incubation with hematoxylin for 1 minute.

### 4.3. Evaluation of Immunohistochesmical Staining

Immunohistochemical analysis was performed according to previously described criteria and methods generally accepted by pathologists [23]. Firstly, staining intensity was classed as negative, weak, moderate, or strong, and stained areas were categorized as having <5% tumor cells, 5%–49% tumor cells, or ≥50% tumor cells. For statistical analysis, a dichotomization method was used based on graded intensity and area. A cutoff point was determined to distinguish unequivocal vivid staining from background staining in tumor cells. The cellular location of staining and the criteria used to classify the 25 tested markers as abnormally expressed are listed in Table 5. As most markers showed positive staining in tumor cells, but not in adjacent non-neoplastic endothelial cells (with the exception of endothelial cell markers CD31, CD34, D2-40, and podoplanin), positive expression was considered to be abnormal expression. However, as p16, p27, and β-catenin were positively expressed in background vessels, loss of expression (i.e., negative expression) of these proteins was considered to be abnormal expression.

### 4.4. Statistical Analysis

A chi-square test or Fisher’s exact test (two-sided) was performed to determine differences in expression between markers and disease groups. A nonparametric Mann–Whitney U test or median test was performed to compare Ki-67 labeling index means and the median value of age. All statistical analyses were conducted using IBM SPSS Statistics 21.0 (IBM, Armonk, NY, USA). Hierarchical clustering of the 130 samples was performed based on the expression rates of the 25 markers, excluding HHV8. A standard Pearson correlation was used as the distance of two samples, and complete linkage clustering was used. Clustering analysis and drawing were performed using the Gene Cluster 3.0 open source clustering software and Java TreeView 1.1 (Eisen Lab Software, Berkeley, CA, USA) [24]. The results were considered to be statistically significant when *p*-values were less than 0.05. 

## Figures and Tables

**Figure 1 ijms-20-03142-f001:**
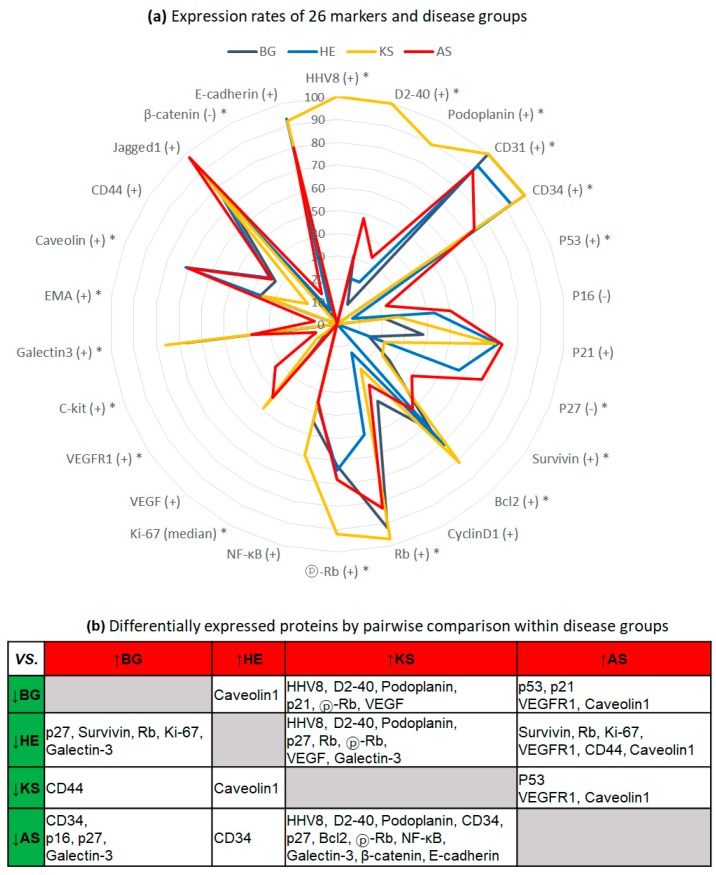
Expression rates of 26 markers according to disease groups. (**a**) Radar chart of expression rates, where the radius is the percentage expression. * *p* < 0.05. (**b**) Differentially expressed proteins obtained by pairwise comparisons between disease groups. Markers that showed significant differences in expression between two groups (chi-square test or Fisher’s exact test *p* < 0.05) are listed in intersecting cells, markers with high expression rates are listed along columns, and markers with low expression rates are listed along rows.

**Figure 2 ijms-20-03142-f002:**
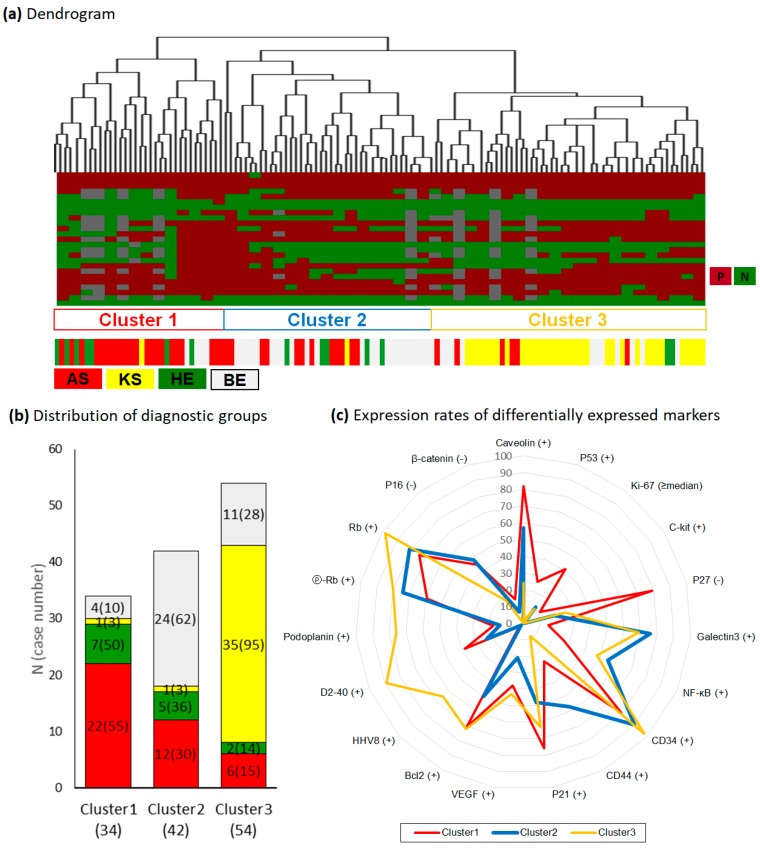
Hierarchical clustering. (**a**) Dendrogram. P: positive, N: negative. (**b**) Distribution of original diagnostic group. N: percentage of case numbers from original diagnostic groups. (**c**) Radar chart of 19 differentially expressed markers within clusters.

**Table 1 ijms-20-03142-t001:** Expression rates of 26 markers in four disease groups.

	N	BG	HE	KS	AS	
N (%)	130	39 (30)	14 (11)	37 (28)	40 (31)	*p*-Value
**Viral Protein**						
HHV8 (+)	37 (28)	0 (0)	0 (0)	37 (100)	0 (0)	<0.001 *
**Endothelial Differentiation**						
CD31 (+)	125 (96)	39 (100)	13 (93)	37 (100)	36 (90)	0.027 *
CD34 (+)	118 (91)	39 (100)	13 (93)	37 (100)	29 (73)	<0.001 *
D2-40 (+)	71 (55)	12 (31)	3 (21)	37 (100)	19 (48)	<0.001 *
Podoplanin (+)	53 (41)	4 (10)	3 (21)	33 (89)	13 (33)	<0.001 *
**Cell Cycle, Apoptosis, or Proliferation**						
P53 (+)	10 (8)	0 (0)	1 (7)	0 (0)	9 (23)	<0.001 *
P16 (−)	44 (34)	8 (21)	6 (43)	10 (27)	20 (50)	0.057
P21 (+)	80 (62)	15 (38)	10 (71)	26 (70)	29 (73)	0.065
P27 (−)	49 (38)	6 (15)	8 (57)	8 (22)	27 (68)	<0.001 *
Survivin (+)	36 (28)	11 (28)	0 (0)	9 (24)	16 (40)	0.036 *
Bcl2 (+)	84 (65)	24 (62)	10 (71)	30 (81)	20 (50)	0.027 *
CyclinD1 (+)	37 (28)	15 (38)	2 (14)	8 (22)	12 (30)	0.26
Rb (+)	112 (86)	36 (92)	7 (50)	36 (97)	33 (83)	<0.001 *
ⓟ-Rb (+)	94 (72)	24 (62)	9 (64)	34 (92)	27 (68)	0.009 *
NF-κB (+)	58 (45)	17 (44)	5 (36)	22 (59)	14 (35)	0.072
Ki-67 (mean ± SD)	23.3 ± 23.7	23.2 ± 22.1	5.0 ± 5.9	19.5 ± 18.3	33.4 ± 28.5	0.001*
**Growth Factor or Protein Kinase**						
VEGF (+)	45 (35)	8 (21)	2 (14)	18 (49)	17 (43)	0.063
VEGFR1 (+)	19 (15)	2 (5)	0 (0)	4 (11)	13 (33)	0.003 *
C-kit (+)	4 (3)	0 (0)	0 (0)	0 (0)	4 (10)	0.026 *
**Cell Adhesion and Motility**						
Galectin-3 (+)	74 (57)	26 (67)	5 (36)	28 (76)	15 (38)	<0.001 *
EMA (+)	4 (3)	0 (0)	0 (0)	0 (0)	4 (10)	0.027 *
Caveolin1 (+)	65 (50)	14 (36)	10 (71)	13 (35)	28 (70)	0.005 *
CD44 (+)	38 (29)	13 (33)	5 (36)	6 (16)	14 (35)	0.086
Jagged1 (+)	112 (86)	28 (72)	11 (79)	34 (92)	39 (98)	0.063
β-catenin (−)	8 (6)	1 (3)	1 (7)	0 (0)	6 (15)	0.033 *
E-cadherin (+)	108 (83)	29 (74)	13 (93)	34 (92)	32 (80)	0.095

ⓟ-Rb, phosphorylated Rb; BG, benign vascular lesion; HE, hemangioendothelioma; KS, Kaposi sarcoma; AS, angiosarcoma; * *p*-value < 0.05.

**Table 2 ijms-20-03142-t002:** Differentially expressed 19 markers among clusters.

	N	Cluster 1	Cluster 2	Cluster 3	
N (%)	130	34 (26)	42 (32)	54 (42)	*p*-Value
Caveolin-1 (+)	65 (50)	28 (82) **^,†,‡^	24 (57)	13 (24)	<0.001 *
P53 (+)	10 (8)	9 (26) **^,‡^	0 (0)	1 (2)	<0.001 *
Ki-67 (≥median)	25 (19)	14 (41) ^**,‡^	5 (12)	6 (11)	0.001 *
C-kit (+)	4 (3)	4 (12) **^,‡^	0 (0)	0 (0)	0.003 *
P27 (−)	49 (38)	27 (79) **^,‡^	8 (19)	14 (26)	<0.001 *
Galectin-3 (+)	74 (57)	5 (15) **^,‡^	32 (76)	37 (69)	<0.001 *
NF-κB (+)	58 (45)	9 (26) **^,‡^	23 (55)	26 (48)	0.013 *
CD34 (+)	118 (91)	27 (79) ^‡^	38 (90)	53 (98)	0.029 *
CD44 (+)	38 (29)	9 (26)	24 (57) **^,†,‡^	5 (9)	<0.001 *
P21 (+)	80 (62)	26 (76)	20 (48) **^,†^	34 (63)	0.015 *
VEGF (+)	45 (35)	13 (38)	9 (21) ^†^	23 (43)	0.025 *
Bcl2 (+)	84 (65)	24 (71)	21 (50) **^,†^	39 (72)	0.044 *
HHV8 (+)	37 (28)	1 (3)	1 (2)	35 (65) ^†,‡^	<0.001 *
D2-40 (+)	71 (55)	13 (38)	10 (24)	48 (89) ^†,‡^	<0.001 *
Podoplanin (+)	53 (41)	6 (18)	6 (14)	41 (76) ^†,‡^	<0.001 *
ⓟ-Rb (+)	94 (72)	20 (59)	31 (74)	43 (80) ^†,‡^	0.001 *
Rb (+)	112 (86)	25 (74)	34 (81)	53 (98) ^†,‡^	0.002 *
P16 (−)	44 (34)	15 (44)	20 (48)	9 (17) ^†,‡^	0.004 *
β-catenin (−)	8 (6)	5 (15)	3 (7)	0 (0) ^†,‡^	0.019 *

ⓟ-Rb, phosphorylated Rb; * among all clusters, *p-value* < 0.05; ** cluster 1 vs. cluster 2; † cluster 2 vs. cluster 3; ‡ cluster 1 vs. cluster 3, *p-value* < 0.05.

**Table 3 ijms-20-03142-t003:** Detailed diagnosis of disease groups (*n* = 130).

Disease Groups	Numbers
**Benign Vascular Lesion (BG)**	39
Acroangiodermatitis	1
Angiofibroma	2
Capillary hemangioma	10
Cavernous hemangioma	3
Cherry angioma	1
Lobular capillary hemangioma	18
Hemangioma	1
Intravascular histiocytosis	1
Stasis dermatitis	2
**Hemangioendothelioma (HE)**	14
Kaposiform hemangioendothelioma	9
Spindle cell hemangioendothelioma	5
**Kaposi sarcoma (KS)**	37
Patch stage	6
Plaque stage	12
Nodular stage	19
**Angiosarcoma (AS)**	40

**Table 4 ijms-20-03142-t004:** Demography of 107 patients according to disease groups.

	Total	BG	HE	KS	AS	
N (%)	107	39 (36)	11 (10)	29 (27)	28 (26)	*p*-Value
Sex						
Male	69 (64)	21 (54)	7 (64)	25 (86)	16 (57)	0.036
Female	38 (36)	18 (46)	4 (36)	4 (14)	12 (43)	
Age	62	58	4.5	68	66	<0.001 *
years, median (min–max)	(2–90)	(2–88)	(7–78)	(24–90)	(7–90)	
Sampling method						
Biopsy	74 (69)	34 (87)	6 (55)	28 (97)	6 (21)	<0.001 *
Resection	33 (31)	5 (13)	5 (45)	1 (3)	22 (79)	
Organ						
Skin	82 (77)	39 (100)	6 (55)	25 (86)	13 (46)	<0.001 *
Non-skin	25 (23)	0 (0)	5 (45)	4 (14)	15 (54)	
Skull	3	0	2	0	1	
Breast	3	0	0	0	3	
Heart	2	0	0	0	2	
Intestine	1	0	0	0	1	
Liver	1	0	0	0	1	
Lung	2	0	0	1	1	
Lymph node	3	0	0	0	3	
Oral cavity and tonsil	3	0	0	3	0	
Pancreas	1	0	1	0	0	
Soft tissue	5	0	1	0	4	
Bone	1	0	1	0	0	
Immune status						
Competent	78 (73)	35 (90)	9 (82)	9 (31)	25 (89)	<0.001 *
Compromised	29 (27)	4 (10)	2 (18)	20 (69)	3 (11)	
HIV infection	0	0	0	4	0	
Chronic disease †	0	0	0	5	2	
Immunosuppressant user	0	0	1	8	0	
Diabetes mellitus	0	4	1	2	1	
Others ‡	0	0	0	1	0	
Disease progression						
No progress	61 (57)	37 (95)	6 (55)	11 (38)	7 (25)	<0.001 *
Progress or deceased	27 (25)	1 (3)	3 (27)	7 (24)	16 (57)	
Censored	19 (18)	1 (3)	2 (18)	11 (38)	5 (18)	

BG, benign vascular lesion; HE, hemangioendothelioma; KS, Kaposi sarcoma; AS, angiosarcoma; * *p-value* < 0.05; † hepatitis C viral hepatitis, chronic renal failure, tuberculosis; ‡ rheumatoid arthritis.

**Table 5 ijms-20-03142-t005:** Information of 25 primary antibodies and interpretation criteria.

Name	Antibody Type (clone)	Dilution	Company (Cat. No.)	Cellular Location	Criteria of Abnormality
HHV8 (LANA-1)	Mouse monoclonal (13B10)	1:100	Cell Marque (CMC885)	N	≥1+, ≥2
CD31	Mouse monoclonal (JC70A)	1:100	DAKO (M0823)	C	≥1+, ≥5%
CD34	Mouse monoclonal (QBEnd-10)	1:300	DAKO (M7165)	C	≥1+, ≥5%
D2-40	Mouse monoclonal (D2-40)	1:100	DAKO (M3619)	C ± M	≥1+, ≥5%
Podoplanin	Mouse monoclonal (unspecified)	1:100	Angiobio Co. (11-003)	C ± M	≥1+, ≥5%
P53	Mouse monoclonal (DO-7)	1:200	DAKO (M7001)	N	≥2+, ≥5%
P16	Mouse monoclonal (G175-405)	1:100	BD Biosciences (551154)	C ± N	≤1+, ≥95%
P21	Mouse monoclonal (187)	1:200	Santa Cruz (sc-817)	N ± C	≥ 2+, ≥5%
P27	Rabbit polyclonal (unspecified)	1:100	Spring Bioscience (E2604)	N	≤1+, ≥95%
Survivin	Rabbit polyclonal (unspecified)	1:500	R&D Systems (AF886)	N ± C	≥1+, ≥5%
Bcl2	Mouse monoclonal (124)	1:100	DAKO (M0887)	N ± C	≥1+, ≥5%
CyclinD1	Mouse monoclonal (DCS-6)	1:100	Santa Cruz (sc-20044)	N	≥1+, ≥5%
Rb	Mouse monoclonal (G3-245)	1:200	BD Biosciences (554136)	N	≥1+, ≥5%
ⓟ-Rb	Rabbit polyclonal (Ser608)	1:30	Cell Signaling (#2181)	N	≥1+, ≥5%
NF-κB	Rabbit polyclonal (unspecified)	1:200	Abcam (ab7970)	C	≥1+, ≥5%
Ki-67	Mouse monoclonal (MIB-1)	1:100	DAKO (M7240)	N	≥1+, ≥5%
VEGF	Rabbit polyclonal (C-1)	1:500	Santa Cruz (sc-7269)	C	≥2+, ≥5%
VEGFR1 (FLT1)	Rabbit polyclonal (C-17)	1:50	Santa Cruz (sc-316)	C	≥2+, ≥5%
C-kit	Rabbit polyclonal (unspecified)	1:200	DAKO (A4502)	C ± M	≥1+, ≥5%
Galectin-3	Mouse monoclonal (9C4)	1:300	Novocastra (NCL-GAL3)	C	≥1+, ≥5%
EMA	Mouse monoclonal (E29)	1:200	DAKO (M0613)	M	≥1+, ≥5%
Caveolin-1	Mouse monoclonal (pY14)	1:300	BD Biosciences (C37120)	C ± M	≥2+, ≥5%
CD44	Mouse monoclonal (DF1485)	1:200	Leica Microsystems (NCL-CD44-2)	M ± C	≥1+, ≥5%
Jagged1	Rabbit polyclonal (H-114)	1:50	Santa Cruz (sc-8303)	C ± M	≥1+, ≥5%
β-catenin	Mouse monoclonal (14/Beta-Catenin)	1:800	BD Transduction (610153)	C ± M	≤1+, ≥95%
E-cadherin	Mouse monoclonal (36/E-Cadherin)	1:500	BD Biosciences (610181)	N	≥1+, ≥5%

ⓟ-Rb, phosphorylated Rb; N, nucleus; C, cytoplasm; M, membrane;. 0, negative; 1+, weak; 2+, moderate; 3+ strong staining.

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
