# Peer review of "Tumor-Associated Protein Profiles in Kaposi Sarcoma and Mimicking Vascular Tumors, and Their Pathological Implications"

_ijms, 2019, doi:10.3390/ijms20133142_

Round 1
Reviewer 1 Report
The authors reported that tumor-associated protein profiles in Kaposi sarcoma and other vascular tumors. They try to distinguish them using protein profiles.
1) In the introduction, the authors reported, "vascular tumors in skin and soft tissue are generally classified into 4 categories." But, you also use WHO classification in soft tissue and bone in the discussion.
2) I think you may exclude the specific sites such as heart, lung and so on. They are the same diagnosis but, tumor characteristics may be different from the skin and soft tissue tumors.
3) I don't understand which groups there were significant differences in Table.1. For example, I don't think there was a statistical difference in CD31 among 4 groups.
4) I don't know why Ki-67 index was high in BG in Table.3. Some tumors may have a high expression of Ki-67 index in benign vascular tumors. But median levels of Ki-67 index should be high in the present study. Which tumors was the high expression of Ki-67 index?
5) In table.4, the median age of 4.5 in HE should be incorrect.
6) In KS patients, 11 patients have died. What was the cause of death?(Kaposi related death?)
Author Response
1) In the introduction, the authors reported, "vascular tumors in skin and soft tissue are generally classified into 4 categories." But, you also use WHO classification in soft tissue and bone in the discussion.
- The WHO classification of skin tumors published in 2013 described vascular tumors as soft tissue tumors of the skin and did not include all vascular lesions, such as Kaposi sarcoma or hemangioendothelioma. However, the 2018 WHO classification of skin tumors included vascular tumors under soft tissue tumors of the skin, following the disease scheme and international classification of diseases for oncology (ICD-O) of soft tissue. Therefore, we cited the WHO classification of vascular tumors in soft tissue, bone, and added it to the reference (page 7, lines 146-148 and Reference No. 4)
The 2018 WHO classification of skin tumors included vascular tumors under soft tissue tumors of the skin, following the disease scheme and international classification of diseases for oncology (ICD-O) of soft tissue [4]
2) I think you may exclude the specific sites such as heart, lung and so on. They are the same diagnosis but, tumor characteristics may be different from the skin and soft tissue tumors.
- As mentioned in Section 4.1 “Patients” of Materials and Methods (page 9, lines 234-236), benign vascular lesions (BG) were collected from skin lesions, whereas hemangioendothelioma (HE), Kaposi sarcoma (KS), and angiosarcoma (AS) patients were selected with no site limitation due to low incidence rates in skin, compared to those of BG. Considering the reviewer’s comment, we have compared protein expression and clustering between skin and non-skin lesions in each disease group; the results have been provided in the attached Supplemental Figure 1 and described in the discussion (page 8, lines 213–220) as follows:
Interestingly, the cluster pattern and protein expression differed according to tumor location (Supplementary Figure 1). Five cases of AS in KS-rich cluster 3 were skin lesions and correlated with high expression of D2-40 and podoplanin, suggesting a lymphatic phenotype. However, NF-κB and galectin-3 were more frequently expressed in skin HE than in non-skin HE. Calveolin levels and loss of p16 were lower in skin lesions than in non-skin lesions. These results confirmed the biologic differences between skin AS and non-skin AS and suggested the biologic heterogeneity of HE according to location.
Supplementary Figure 1.
3) I don't understand which groups there were significant differences in Table.1. For example, I don't think there was a statistical difference in CD31 among 4 groups.
- CD31 was highly expressed in all groups. However, the four CD31-negative cases were all AS. This factor influenced the total analysis. The results of pairwise comparison within groups were as follows:
Total | BG | HE | KS | AS | p-value | |
CD31 (+) [N(%)] | 125(96) | 39(100) | 13(93) | 37(100) | 36(90) | 0.027* |
Person chi-square p-value | HE | NA | ||||
KS | NA | NA | ||||
AS | 0.048* | 0.236 | 0.048* |
4) I don't know why Ki-67 index was high in BG in Table.3. Some tumors may have a high expression of Ki-67 index in benign vascular tumors. But median levels of Ki-67 index should be high in the present study. Which tumors was the high expression of Ki-67 index?
- We have described in the Results (page 5, lines 90–95) as follows:
Benign vascular lesions with Ki-67 expression exceeding the median value were capillary hemangioma (3/10) and granuloma pyogenicum (12/18); Ki-67 expression levels in these lesions were 46.33±16.84% (mean±SD). BG lesions with Ki-67 expression below the median value included acroangiodermatitis, angiofibroma, cavernous hemangioma, cherry angioma, hemangioma, intravascular histiocytosis, and stasis dermatitis; Ki-67 expression levels were 8.71±8.10% (mean±SD).
5) In table.4, the median age of 4.5 in HE should be incorrect.
- Eight patients with kaposiform HE in the HE group were children under 4 years old. Therefore, the median age of 4.5 years is correct.
6) In KS patients, 11 patients have died. What was the cause of death? (Kaposi related death?)
- Seven (not 11) of the patients with KS listed in Table 4 showed disease progression or death. One patient died owing to hemorrhage of Kaposi sarcoma in the upper airway, and the remaining six patients survived with disease progression.
Reviewer 2 Report
PROS: The design of the manuscript is kept simple with limited methods used and is written well.
CONS:
1. What is the significance of elevated D2-40 and podoplanin in KS alone compared to other cancer conditions like AS and HE?
2. How does downregulation of CD44 aid in KS pathogenesis?
3. Expression of the key proteins should also be confirmed by qRT-PCR. This will provide information regarding the rate of transcription of these genes as well.
4. So what is the model describing the pathology based on the protein profiling?
Author Response
1) What is the significance of elevated D2-40 and podoplanin in KS alone compared to other cancer conditions like AS and HE?
As mentioned in the Discussion (page 7, line 166-168), HHV8-infected endothelial cells are reprogrammed to lymphatic endothelial cells by induction of transcription factors such as Prox-1, which is a key regulator of lymphatic differentiation. Some AS and HE can also express D2-40 or podoplanin; thus, evidence of lymphatic differentiation is insufficient for diagnosis of KS, and the co-presence of HHV8 infection is more specific evidence of KS.
2) How does downregulation of CD44 aid in KS pathogenesis?
It is unclear whether CD44 is downregulated in KS pathogenesis or not, we have mentioned the hypothesis in the discussion and added references (page 8, line 202-207 and reference No. 21 22).
CD44 is a blood vascular endothelium-specific hyaluronan receptor. It is unclear whether CD44 is downregulated in KS during virus-induced lymphatic programing or KS arises from HHV8-infected lymphatic endothelial cells (LEC) not expressing CD44. Another hypothesis is that stem cells are involved in the pathogenesis of KS. CD44 is one of several surface markers for mesenchymal stem cells, and HHV8-infected mesenchymal stem cells show a loss of stem cell markers
3) Expression of the key proteins should also be confirmed by qRT-PCR. This will provide information regarding the rate of transcription of these genes as well.
Underlying genetic abnormalities may be good molecular evidence of altered protein expression. However, our study was not focused on genetic alterations but rather protein expression profiles to better understand the vascular lesions histologically mimicking Kaposi sarcoma.
4) So what is the model describing the pathology based on the protein profiling?
We have clarified this model in the conclusion (page 8, lines 225-229).
Vascular tumors mimicking Kaposi sarcoma can be categorized into three clusters: cluster 1, altered expression of tumor suppressor genes (p53, p27) with a high proliferative index, enriched AS; cluster 2, high CD44 expression with a low proliferative index suggesting stem cell differentiation, mixed diagnosis; and cluster 3: lymphatic differentiation with activated Rb pathway, enriched KS.
Round 2
Reviewer 1 Report
The authors added the results and discussions appropriately. No additional comments.
Reviewer 2 Report
ACCEPT!
Congratulations.